# Analysis of Multiple Drug Resistance Mechanism in Different Types of Soft Tissue Sarcomas: Assessment of the Expression of ABC-Transporters, MVP, YB-1, and Analysis of Their Correlation with Chemosensitivity of Cancer Cells

**DOI:** 10.3390/ijms23063183

**Published:** 2022-03-16

**Authors:** Natalia I. Moiseeva, Lidia A. Laletina, Timur I. Fetisov, Leyla F. Makhmudova, Angelika E. Manikaylo, Liliya Y. Fomina, Denis A. Burov, Ekaterina A. Lesovaya, Beniamin Y. Bokhyan, Victoria Y. Zinovieva, Alice S. Vilkova, Larisa V. Mekheda, Nikolay A. Kozlov, Alexander M. Scherbakov, Evgeny M. Kirilin, Gennady A. Belitsky, Marianna G. Yakubovskaya, Kirill I. Kirsanov

**Affiliations:** 1N.N. Blokhin National Medical Research Center of Oncology, 115478 Moscow, Russia; panlidia@gmail.com (L.A.L.); timkatryam@yandex.ru (T.I.F.); mahmusha@yandex.ru (L.F.M.); 7717271@mail.ru (A.E.M.); 3050244@gmail.com (L.Y.F.); denisburov@yandex.ru (D.A.B.); lesovenok@yandex.ru (E.A.L.); beniamin-bokhyan@mail.ru (B.Y.B.); vichka2396@gmail.com (V.Y.Z.); cunning.fox@mail.ru (A.S.V.); lmeheda@gmail.com (L.V.M.); newbox13@mail.ru (N.A.K.); alex.scherbakov@gmail.com (A.M.S.); belitsga@mail.ru (G.A.B.); mgyakubovskaya@mail.ru (M.G.Y.); kkirsanov85@yandex.ru (K.I.K.); 2Department of Oncology, I.P. Pavlov Ryazan State Medical University, 9 Vysokovoltnaya St., 390026 Ryazan, Russia; 3Belozersky Institute of Physico-Chemical Biology, Lomonosov Moscow State University, 119991 Moscow, Russia; kirilin@belozersky.msu.ru; 4Institute of Medicine, RUDN University, 117198 Moscow, Russia

**Keywords:** soft tissue sarcoma, chemosensitivity test, ABCB1, ABCC1, ABCG2, MVP, YB-1, Pgp, undifferentiated pleomorphic sarcoma, synovial sarcoma

## Abstract

Chemotherapy of soft tissue sarcomas (STS) is restricted by low chemosensitivity and multiple drug resistance (MDR). The purpose of our study was the analysis of MDR mechanism in different types of STS. We assessed the expression of ABC-transporters, *MVP*, *YB-1*, and analyzed their correlation with chemosensitivity of cancer cells. STS specimens were obtained from 70 patients without metastatic disease (2018–2020). Expression level of MDR-associated genes was estimated by qRT-PCR and cytofluorimetry. Mutations in ABC-transporter genes were captured by exome sequencing. Chemosensitivity (SI) of STS to doxorubicin (Dox), ifosfamide (Ifo), gemcitabine (Gem), and docetaxel (Doc) was analyzed in vitro. We found strong correlation in *ABCB1*, *ABCC1*, and *ABCG2* expression. We demonstrated strong negative correlations in *ABCB1* and *ABCG2* expression with SI (Doc) and SI (Doc + Gem), and positive correlation of *MVP* expression with SI (Doc) and SI (Doc + Gem) in undifferentiated pleomorphic sarcoma. Pgp expression was shown in 5 out of 44 STS samples with prevalence of synovial sarcoma relapses and it is strongly correlated with SI (Gem). Mutations in MDR-associated genes were rarely found. Overall, STS demonstrated high heterogeneity in chemosensitivity that makes reasonable in vitro chemosensitivity testing to improve personalized STS therapy, and classic ABC-transporters are not obviously involved in MDR appearance.

## 1. Introduction

Success in chemotherapy of soft tissue sarcomas (STS) is still difficult to achieve due to multiple obstacles. STS are heterogeneous cancers with more than 100 histological subtypes, different in molecular abnormality profiles [1] that make investigations in the field of STS personalized therapy very complex [2]. In spite of the multiple ongoing clinical trials for novel targeted drugs [3], no one of them was introduced in clinical practice at the first line of treatment, and combination of doxorubicin and ifosfamide remain the gold standards chemotherapy [4,5], as well as gemcitabine and docetaxel are often used in second line. Unfortunately, chemotherapy is efficient for not more than 50% of patients and it is followed by fast development of MDR. In different other cancer types these drugs usually promote the development of MDR through the activation of ABC-transporters [6].

ABC-transporters play a key role in cell viability by transporting essential molecules, toxins, xenobiotics and metabolites inside and outside the cell [7]. One of the most important ABC-transporters for MDR development is P-glycoprotein (Pgp, ABCB1/MDR1). Both doxorubicin and docetaxel represent the substrates for Pgp [8]. ABCC1 (MRP1) and ABCG2 (BCRP) are other members from ABC-transporters family [7]. In breast cancer high level of ABCG2 expression is observed in stem cancer cells [9]. The role of mutations in ABC-transporter genes on the protein function has become a new object of interest, which is under the intensive investigation [10].

Drug reflux-associated MVP/LRP (major vault protein/lung resistance protein) also plays an important role in MDR. MVP/LRP is the principal component of vaults, which are the largest ribonucleoprotein particles with barrel-like structures. These vaults may mediate MDR via drug sequestration and exocytosis [11].

Activation of several ABC-transporters is induced by transcription factors from a number of signaling pathways. One of these transcription factors is DNA/RNA-binding protein YB-1 [12], which regulate the expression of Pgp [13], MVP [14], and possibly MRP1. Overexpression of YB-1 in tumors is often associated with highly malignant phenotype and poor prognosis [15].

Contribution of ABC-transporters to MDR in STS was studied earlier. In particular, ABCB1 and ABCC3 activation and an increase in corresponding protein levels associated with pour prognosis were shown in malignant peripheral nerve sheath tumor (MPNST) [16]. Moreover, Pgp expression was shown to correlate significantly with large tumor size and high AJCC stages (III and IV) [16]. Other study of STS prognostic factors revealed the correlation of Pgp expression with poor outcome [17]. For other types of STS the results were controversial.

The main goal of our study was the analysis of MDR mechanism in different types of STS: undifferentiated pleomorphic sarcoma, liposarcoma, synovial sarcoma, leiomyosarcoma, and others.

## 2. Results

### 2.1. Evaluation of the Chemosensitivity of Primary Cultured STS Cells to Anticancer Drugs

We obtained 70 primary cell cultures from STS samples. At passages 0 and 2–3 we evaluated the percentage of tumor cells in the cultures with cytological staining. In further analysis we used the cultures with ≥90% of tumor cells (Figure 1).

We evaluated the resistance of cancer cells from STS primary cultures to doxorubicin (Dox), ifosfamide (Ifo), docetaxel (Doc), gemcitabine (Gem), and the combinations: (Dox + Ifo) and (Doc + Gem). If the sensitivity index (SI) was less than 300, cells were accounted as sensitive to the drug or drug combination. Figure 2 demonstrates the histograms of SI frequencies for all drugs individually and for drug combinations. It is noteworthy that primary cells revealed the high heterogeneity; therefore, the SI frequencies did not match to normal distribution. Resistant STS are more frequently occurred in case of Doc and Gem: SI > 300 in 74.3% and 67.1% of samples after individual treatment and 47.1% after combination treatment. In case of Dox and Ifo these indexes were 51.4% and 46.7%, respectively, and 25.9% for combination.

SIs of studied drugs demonstrated moderate positive correlation among themselves (r interval was 0.44–0.72, *p* < 0.0001). Strong correlation was shown for SI (Doc) and SI (Gem) with SI (Doc + Gem), r = 0.85 и r = 0.84, *p* < 0.0001, respectively, as well as between SI (Ifo) and SI (Dox + Ifo), r = 0.83, *p* < 0.0001.

STS cells demonstrated higher resistance to Doc and Gem, than to Dox and Ifo (*p* < 0.001). Drug combinations were more effective than individual treatments (*р* < 0.01), and combination Dox + Ifo showed higher cytotoxicity than Doc + Gem (*р* = 0.03) (Table 1). Forty-one percent of primary cell cultures were sensitive to both drug combinations, and 22% of them showed resistant phenotype.

Furthermore, we observed higher resistance to Dox in tumor cells from the patients of the age above 40 years old (*р* = 0.01). The SI of other drugs were not associated with age, sex, tumor size and previously applied chemotherapy. We compared the SI level in undifferentiated pleomorphic sarcoma, liposarcoma, and synovial sarcoma as these groups contained the largest number of samples. Undifferentiated pleomorphic sarcoma cells were more resistance to Dox then synovial sarcoma (*р* = 0.01 for pairwise comparison and *р* = 0.04 for group comparison). We did not observe any correlation between SI of other drugs and tumor histology type. However, we showed an interesting association between SI (Doc) and tumor regression grade: groups of patients, who received adjuvant or neoadjuvant chemotherapy and demonstrated tumor regression grade of 2–3 level, have Doc-resistant tumor cells more frequently: SI (Doc) 304 ± 122 in group with 0–1 tumor regression grade vs. SI (Doc) 487 ± 74 in group with 2–3 tumor regression grade (*р* = 0.0014). Moreover, comparing the patient groups with neoadjuvant therapy only, SI (Doc), and SI (Doc + Gem) were higher in primary cell cultures from the patients with better response to the therapy (*р* = 0.008 and *р* = 0.03, respectively) (Table 2).

### 2.2. Expression of MDR-Associated Genes in STS

We analyzed the relative expression of ABC-transporters’ genes *ABCB1*, *ABCC1*, and *ABCG2*, transcription factor *YB-1* and *MVP* gene associated with the distribution of xenobiotics in cytoplasm, in 70 tumor samples. We revealed the strong positive correlation between *ABCB1*/*ABCC1* and *ABCB1*/*ABCG2* expression and between *ABCC1*/*ABCG2* expression. *MVP* expression demonstrated weak positive correlation with *YB-1* and weak negative correlation with *ABCB1*. *YB-1* expression did not correlate with the expression of ABC-transporters (Table 3).

We observed higher *ABCG2* expression level (*р* = 0.007) in patients at the age above 40 years old. Expression of other studied genes were not associated with age, sex, tumor size, degree of malignancies and tumor regression grade (Table 4). In case of undifferentiated pleomorphic sarcoma *YB-1* expression was higher than in liposarcoma samples (*р* = 0.009 for pairwise comparison, *р* = 0.03 for group comparison). In relapsed patients we observed the decreased expression of *MVP* (*р* = 0.04).

Generally, expression of studied genes did not correlate with the drug resistance (SI), except for weak negative correlation for *ABCB1* expression and SI (Doc), r = −0.26; *p* = 0.02. *MVP* expression demonstrated weak positive correlation with SI (Doc), r = 0.24; *p* = 0.04.

Then we analyzed the correlation between expression of MDR-associated genes and SIs of chemotherapeutics in three groups of STS of different histological subtypes: undifferentiated pleomorphic sarcoma (24 samples), liposarcomas (16 samples), synovial sarcomas (11 samples). In liposarcomas we did not show any correlations in SIs of studied drugs. In undifferentiated pleomorphic sarcoma we demonstrated the medium negative correlation of *ABCB1* and *ABCG2* expression with SI (Doc) and SI (Doc + Gem). In further analysis we showed that these correlations were more pronounced in the group of undifferentiated pleomorphic sarcoma relapses. Moreover, we found the positive correlation of *MVP* expression with SI (Doc) and SI (Doc + Gem) in undifferentiated pleomorphic sarcoma samples from patients treated with neoadjuvant chemotherapy (Table 5). In synovial sarcomas we demonstrated positive correlation of *ABCC1* expression with SI (Ifo) and SI (Dox + Ifo) as well as *ABCG2* expression gene with SI (Doс). We did not analyze the groups of primary and relapsed patients because of insufficient groups size. 

### 2.3. Expression of P-Glycoprotein and ABCG2 in Primary Cultures of STS

We analyzed the protein Pgp and ABCG2 expression in primary STS cultures of 1–2 passages by flow cytometry. We used cell cultures K562/i-S9 with Pgp overexpression and HBL-100/Dox with ABCG2 overexpression as positive controls.

Pgp expression was evaluated in 44 samples. The threshold value was 5% of cells population, which expressed Pgp. We supposed, it was the minimal level of protein expression with significant contribution in the resistance of tumor population and with the effect on clonal selection under the drug exposure. In 5 samples (11%) the Pgp expression was above this value with mean expression level 19.4% of STS cell population in positive group (Figure 3). As in this sample series the amount of medium and high Pgp expression were lower than expected, we selected 8 STS samples with different Pgp expression and analyzed the protein amount by Western blotting (Figure 4). The results of flow cytometry were confirmed by Western blot analysis that supported the observation on rare Pgp expression in studied sample series.

ABCG2 expression was analyzed in 39 samples, the threshold value was 1% as ABCG2 expresses in cancer stem cells being the minor population. We found 17 samples (44%) with the expression level above this threshold with mean expression level 3,75% of STS cell population in positive group (Figure 5). Thus, we did not find correlations in *ABCB1* and *ABCG2* mRNA expression and Pgp and ABCG2 protein expression, respectively.

Pgp and ABCG2 expressions were not associated with such clinical patient characteristics as sex, age, disease stage, degree of malignancy, chemo/radiotherapy delivery and the treatment response (Table 6). ABCG2 expression did not vary in different histological STS subtypes. Within the subgroups we did not find the association of ABCG2 with SI of primary STS cultures.

We demonstrated that Pgp expression frequency in synovial sarcomas was significantly higher than in undifferentiated pleomorphic sarcomas (*p* = 0.04). In synovial sarcomas 5 out of 6 samples were obtained from the relapsed patient, and in this subgroup, we showed strong positive correlation of Pgp expression with SI (Gem) (r = 0.97, *p* = 0.005). However, it should be mentioned that Gem is not Pgp substrate.

### 2.4. Identification of Mutations in Several Genes of ABC-Transporter Family

We analyzed the mutations in several genes from ABC-transporter family in the group of 39 STS samples (synovial and pleomorphic histology subtypes). We selected 36 genes associated with tumor progression out of total 300 genes from ABC-transporters family. We found mutations in 9 out of 39 samples, in 3 samples (STS 29, 104, 119) we observed 2–3 mutations. In the Table 7, A3, A13, C1, C6, C11, B1, and G2 genes are associated with drug elimination from cell, whereas A1 and B11 genes play a role in transport of lipids and biliary acids. Mutations were found in the most significant MDR-associated genes B1, С1, G2 in 1, 2, and 1 cases, respectively (10% of total number of samples).

## 3. Discussion

We hypothesized that expression of ABC-transporters is the important mechanism of MDR in STS as it was shown for malignant peripheral nerve sheath tumor and sarcomas of the pulmonary artery [16,18]. Further, it was described earlier that *ABCC1*/*MRP1* expression may have a prognostic significance for patients with STS of high risk, who received anthracycline-based treatment [19]. In cases of embryonal rhabdomyosarcoma *ABCC1* expression correlated with poor survival [20]. More frequently this disease is associated with ABCG2 expression [21].

However, in our study we found only 11% of samples with Pgp expression, and only in 1 case we observed 30% Pgp-positive cells. The similar results were obtained for ABCG2 expression: This index reached 16% in rare cases and generally did not exceed 2–5%. As the low level of Pgp and ABCG2 expression in cancer cells resistant to the studied drugs was observed, we propose that other mechanisms of MDR may be more important in STS. Possibly, the rare Pgp expression is associated with the histological tumor subtypes in our specific sample series as in literature high Pgp expression was described in rhabdomyosarcomas [17] and low expression of this gene was shown to be associated with Ewing sarcoma [22]. Interesting, results were obtained for the subgroup of 6 samples of synovial sarcoma, 5 of which belonged to relapsed patients. We demonstrated the increased Pgp expression strongly positively correlated with the resistance to Gem, nevertheless, Gem is not the Pgp substrate. However, one study described the increased sensitivity to Gem in Pgp or MRP1-overexpressing cell lines and their counterparts without MDR phenotype [23]. In our study, we suppose, that Pgp expression does not lead to Gem resistance, but Gem may activate Pgp expressions [24].

In primary analysis of the whole sample series, we did not observe any strong correlations of MDR-associated genes expression and resistance of primary cell cultures. Moreover, in the analysis of undifferentiated pleomorphic sarcoma subgroup we demonstrated the negative correlation for the *ABCB1* and *ABCG2* expression and SI (Doc) or SI (Gem + Doc). As corresponding proteins were not expressed on the sufficient level in STS cells in our sample series, we guess that decreased expression of *ABCB1* and *ABCG2* in resistant cells is the indicator of the activation of the specific signaling resulting in STS resistance. If we consider other MDR mechanisms in undifferentiated pleomorphic sarcoma, then it may be the activation of LAPTM4A and LAPTM4B, which are responsible for the transport of small molecules through endosomal and lysosomal membranes. Its increased expression correlated with lower chemosensitivity, especially to the anthracycline-based therapy [25].

We found that studied ABC-transporter genes correlated with each other by mRNA expression level; however, we did not observe sufficient protein amount in spite of high level of mRNA expression in most samples. We believe that it could be associated with the alternative regulation on translation level. It is known that small non-coding RNA (miRNA) participate in posttranscriptional regulation of gene expression either via activation of mRNA degradation or via inhibition of translation repression [26,27]. Thus, canonic mRNA binding with complementary seed regions in 3′UTR mRNA results in translation repression, and, similarly, regulatory RNA-binding proteins change its stability or further translation process [28] 38. In one study it was shown that the three-component complex of HuR, miR-19b, and UTR inhibits the expression of ABCB1/Pgp [29]. In another study it was demonstrated that interactions of ABCB1-3′-UTR—miR-485-3p and ABCC2-3′-UTR—miR-26a-5p induce the translation repression of Pgp and MRP1, which was resulted in the low expression on protein level in spite of the high mRNA expression observed [30].

In our study we demonstrated that resistance to Doc and its combination with Gem was higher in patients with good response to the therapy Dox + Ifo. It could prove the hypothesis that tumor cells survived the neoadjuvant chemotherapy and undergone the clonal selection, obtain higher resistance to Doc. In addition, it could be in accordance with the data that viability of tumor cells with low-grade therapy-related pathomorphism is more dependent from the tumor microenvironment [31]. After culturing in vitro this protective property was withdrawn, and cells demonstrated the chemosensitivity in MTT-test. 

We found that the expression of *MVP (LRP)* gene on mRNA level is lower in tumor samples from the relapsed patients than in primary tumors. It could be associated with the differentiation grade in primary and relapsed tumors. In one study it was demonstrated that MVP protein was predominantly expressed in differentiated cells in rhabdomyosarcoma samples before and after chemotherapy, possibly MVP expression allow rhabdomyosarcoma cells to survive the chemotherapy [32]. In analysis of undifferentiated pleomorphic sarcoma subgroup, we found the positive correlation in *MVP* expression and resistance to Doc and Doc + Gem combination, and this correlation become stronger in the cells obtained from patients after neoadjuvant chemotherapy. That was in agreement with our previous study on glioblastoma samples. We showed that increased *MVP* expression on mRNA level was associated with the low proliferation and high resistance to Temozolomide [33]. In the undifferentiated pleomorphic sarcoma subgroup, we proposed direct action of MVP as an MDR protein, but this conclusion requires further verification on a larger group of patients and study of MVP at the protein level. 

Our study had a number of limitations: Some of the analyzed groups had very few samples. Almost the entire group of synovial sarcomas with increased Pgp expression came from relapsed patients, and we cannot establish whether high Pgp expression is characteristic only of relapse samples or of all synovial sarcomas. The next limitation is that at this stage of investigation, we studied the expression of *YB-1*, *ABCC1*, and *MVP* only at the mRNA level, but not at the protein level, which could provide new information concerning their involvement in the MDR formation of STS.

## 4. Materials and Methods

### 4.1. Methodological Approaches

We assessed the expression of ABC-transporters, *MVP* and *YB-1* and analyzed their correlation with chemosensitivity of corresponding STS cells to the above-mentioned chemotherapeutic agents. Drug resistance was estimated using in vitro chemosensitivity assay proposed by Kurbacher et al. [34], which, currently, are widely used in many modifications [35,36,37,38], based on different methods for estimation of cell viability [39,40,41].

### 4.2. Tumor Specimens

A total of 70 fresh tumor specimens were obtained from patients who had soft tissue sarcoma and underwent surgery at the N.N. Blokhin National Medical Research Center of Oncology in 2018–2020 years. The prospective study included patients over 18 years old with diagnosed soft tissue sarcoma confirmed by histology, stages I–III. Patients with metastatic disease were excluded from the study. The average age of patients was 46.2 ± 16.2 years and there were 37 males and 33 females. Tumor localization was as follows: lower extremity—42 (60.0%), upper extremity—10 (14.3%), trunk—13 (18.6%), head and neck—5 (7.1%). The STS stages were distributed as follows: I—1 (1.4%), II—14 (20.0%), III—54 (77.2%), IV—1 (1.4%). STS histology types included 24 undifferentiated pleomorphic sarcoma (34.2%), 16 liposarcoma (22.9%), 11 synovial sarcoma (15.7%), 6 leiomyosarcoma (8.6%), 4 malignant schwannoma (5.7%), epithelioid sarcoma (5.7%), dermatofibrosarcoma (4.3%) and Ewing sarcoma (2.9%). Liposarcoma was represented by the following types: 9 myxoid liposarcoma (56.3%), 6 dedifferentiated liposarcoma (37.5%), and 1 pleomorphic liposarcoma (6.2%). The studied STS cohort included 37 newly diagnosed patients (52.9%) and 33 patients (47.1%) with recurrent disease. Chemotherapy in the neoadjuvant setting was administered to 13 patients (18.6%), all patients received 2–4 courses of doxorubicin and ifosfamide. Signed written informed consents were obtained from all participants before the study. 

### 4.3. Chemotheraputic Drugs

In our study we used doxorubicin (Dox, RONC, Moscow, Russia), docetaxel (Doc, NATIVA, Moscow region, Russia), and gemcitabine (Gem, BIOCAD, Moscow, Russia). Because ifosfamide (Ifo) is a prodrug and it requires in vivo hepatic activation, we used the active metabolite 4-hydroperoxy-Ifosfamide (4-OH-Ifo, NIOMECH, Bielefeld, Germany).

### 4.4. Primary Cancer Cell Cultures and Chemosensitivity Assay

Tumor chemosensitivity assay was performed as a routine procedure immediately following surgery. Solid tumors were obtained during surgery and cut into smaller fragments (1 mm^3^), which were then dissociated by incubation in 5–10 mL sterile collagenase mix for 2–3 h at 37 °C on a shaker to prepare suspensions of single cells. One part of the cells was used for the chemosensitivity test, and the other one for the analysis of multi-drug resistance. Аfter adjusting the concentration of cells in the suspension to 1–2 × 10^5^ cells/mL, 100-μL cell suspensions were added to each well of a 96-well microplate. Single agents were tested at six different concentrations of a standard test drug concentration (TDC), in particular, 6.25, 12.5, 25, 50, 100, and 200% of the peak plasma concentration of the drug (Table 8), as it was proposed by Andriotti et al. [42].

The TDCs were based on pharmacokinetic data for standard doses of the agents, adjusted to give good discrimination [42]. Plates were incubated for 5–6 days at 37 °C with 95% humidity in a 5% CO_2_ incubator. Cell viability was measurement using resazurin-based assay described previously [25]. The results in vitro chemosensitivity tests were interpreted and compared using the sensitivity index SI (SI = 600—sum of % inhibition at 200, 100, 50, 25, 12.5, and 6.25% TDC) [42,43,44].

### 4.5. Cytology for Isolated Cell Culture

In order to determine the percent of malignant cells for each isolated primary culture, cytological study was performed. Derived cells were used to thin-layer slides preparation through a cytocentrifugation process by the Thermo Shandon Cytospin 3. Morphological assay to determine the percent of malignant cells in Leishman-stained slides were performed on a microscope «Nikon Eclipse Ci-S» (Nikon Corporation, Tokyo, Japan) at 1000× magnification in 3 fields of view.

### 4.6. Quantitative PCR (Q-PCR)

Total RNA was isolated with PureZOL RNA Isolation Reagent (BIO-RAD, Hercules, CA, USA) according to the manufacturer’s protocol. The RNA quality was checked by electrophoresis in 1% agarose gel containing 0.01% ethidium bromide. Samples with clearly visible 18S and 28S RNA bands were used for further analysis. For the synthesis of cDNA, we used a set of reagents for reverse transcription with primers Random6 (Syntol, Moscow, Russia). The real-time PCR reaction was performed using the intercalating fluorescent agent Eva Green (Synthol, Moscow, Russia) and Taq DNA polymerase from ThermoScientific (EP0402), the CFX Connect Real-Time PCR Detection System (BIO-RAD, USA) was used. Amplification steps: 95 °C—3:00 min, (95 °C—0:10 min, 60 °C—00:10 min, 72 °C—00:30 min)—39 cycles, melt curve 65 °C–95 °C. The primer sequences were selected in PrimerBank (PrimerBank. Available online: https://pga.mgh.harvard.edu/primerbank/ (accessed on 20 November 2018). The results were normalized using housekeeping *RPL27* gene. The following pairs of primers were used: *YBX-1* forward CCCCAGGAAGTACCTTCGC, reverse AGCGTCTATAATGGTTACGGTCT; *ABCB1* forward GGGATGGTCAGTGTTGATGGA, reverse GCTATCGTGGTGGCAAACAATA; *ABCC1* forward GTGAATCGTGGCATCGACATA, reverse GCTTGGGACGGAAGGGAATC; *MVP* forward TACATCCGGCAGGACAATGAG, reverse CTGTGCAGTAGTGACGTGGG; *ABCG2* forward TGAGCCTACAACTGGCTTAGA, reverse CCCTGCTTAGACATCCTTTTCAG; *RPL27* forward ACCGCTACCCCCGCAAAGTG, reverse CCCGTCGGGCCTTGCGTTTA.

### 4.7. Flow Cytometry

The expression of Pgp and ABCG2 proteins was assessed in primary cultures of STS by flow cytometry on a BD FACSCanto II flow cytometer with BD FACSDiVa software v6 1.3 (BD Biosciences, Franklin Lakes, NJ, USA). Cells were fixed by 4% paraformaldehyde solution in PBS for 10 min, when cells washed twice in PBS. After suspension of 500 thousand cells per point was incubated in 100 μL of PBS with antibodies in the ratio specified in the manufacturer’s protocol for 40 min in the dark at room temperature, then washed twice in PBS. Fixable Viability Stain 510 (FVS510) was used as a marker of cell viability (Biosciences, Cat. No. 564406). The K562/i-S9 cell line with Pgp overexpression (obtained from K562 by transfection with the *MDR1* (*ABCB1*) gene) was used as a positive control. K562/i-S9 cell line was kindly provided by Mechetner [45]. In our laboratory of tumor cell genetics, we were derived a doxorubicin-resistant cell subline of Pgp and ABCG2 (BCRP) overexpressing HBL-100/Dox cell line from the HBL-100 cells (received from Institute of Cytology RAS) after the continuous cell selection in the presence of Dox [46]. HBL-100/Dox cell was used as a positive control for anti-ABCG2 antibody performance. The range of autofluorescence for the samples varied from 0.1 to 10%. Antibodies were used in the work: FITC-Pgp (BD Biosciences, USA, clone 17F9, Cat. No. 557002), APC anti-human CD338 (ABCG2) (BioLegend, San Diego, CA, USA, clone 5D3, Cat. No. 332020).

### 4.8. Western Blotting

Two million cells were suspended in 300 µL of RIPA buffer x1 (Thermo Scientific, Waltham, MA, USA) for 20–30 min at +4 °С. Lysates were centrifuged at 13,400 rpm for 30 min, supernatants were placed in 4× sample buffer (1 M Tris-HCl, pH 6.8, 10% SDS, 50% glycerol, 10% β-mercaptoethanol, and bromophenol blue) to 1x-dilution. Samples were placed in the water bath for 10 min at 96 °С. Proteins were resolved in 10% PAAG with 10% SDS and transferred on nitrocellulose membrane (Amersham, Chicago, IL, USA). Membranes were blocked in 5% BSA in 1× TBST (Tris buffer pH 7.5 with 0.005% Tween-20) for 1 h at room temperature. Then membranes were incubated with specific primary monoclonal antibodies against Pgp (MDR1/ABCB1 (D3H1Q) Rabbit mAb Cat. No.12683, Cell Signaling, Danvers, MA, USA) at 1:800 dilutions overnight at 4 °С. For the normalization membranes were incubated with anti-actin antibodies (β-Actin Antibody (C4) HRPsc-47778 HRP, Santa Cruz Biotechnology, Dallas, TX, USA), at 1:500 dilutions. Membranes were washed in TBST three times for 10 min and incubated with secondary peroxidase-conjugated antibodies at 1:10,000 dilutions (Jachson ImmunoResearch, Dallas, TX, USA). Then membranes were washed for three times and 10 min in TBST and visualized with ECL reagent (Thermo Fisher, Waltham, MA, USA) and ImageQuant Las 4000 (GE Healthcare, Chicago, IL, USA).

### 4.9. Exome Capture, Alignments and Base-Calling

Exomes for the 39 patients were captured with Agilent SureSelect Focused Exome. Libraries were indexed, pooled, and sequenced on Illumina HiSeq2000 machines (paired-end, 250-bp reads). Reads were mapped to GRCh38/hg38 build using BWA 0.7.15 [47] followed by marking duplicate reads with Picard-tools 2.20 (Picard-tools 2.20. Available online http://broadinstitute.github.io/picard/ (accessed on 14 September 2020)). GATK4 [48] base quality score recalibration pre-processing step was performed to detect systematic errors made by the sequencing machine. Tumor-only variant calling was performed on tumor samples with no paired normal using advantages of normal cell contamination implemented in GATK4 Mutect2 (tumor-only mode) probabilistic models for genotyping and filtering.

### 4.10. Statistical Analysis

Data were presented as M ± S.D. The Mann–Whitney test was carried out to analyze the differences between groups and the Spearman’s correlation coefficient was calculated to quantitate the degree of correlation between parameters. The Kruskal–Wallis test with post test (compare all pairs of columns) was used to analyze groups with different histology types. GraphPad Prism 6.0 was used (GraphPad Software, San Diego, CA, USA). The difference was considered statistically significant at *p*-value < 0.05 (two-tailed).

## 5. Conclusions

Overall, we used STS primary cultures as a new approach for the study of MDR mechanisms in STS of different types. Pgp protein expression is a very rare event and probably does not have clinical significance, with the exception of synovial sarcoma. Our results afford us to conclude that MVP expression may play role in MDR of undifferentiated pleomorphic STS. Mechanisms of resistance in STS require further study and search for new targets.

## Figures and Tables

**Figure 1 ijms-23-03183-f001:**
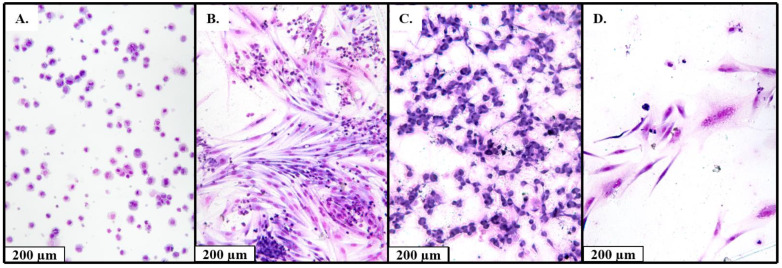
Examples of cytological studies of the derived cell cultures (Leishman stain; 80×). (**A**) Cell culture obtained from the patient with synovial sarcoma (99% malignant cells), (**B**) cell culture obtained from the patient with malignant schwannoma (20% malignant cells, 80% lymphocytes), (**C**) cell culture obtained from the patient with undifferentiated pleomorphic sarcoma (99% malignant cells), (**D**) cell culture obtained from the patient with liposarcoma (99% of malignant cells).

**Figure 2 ijms-23-03183-f002:**
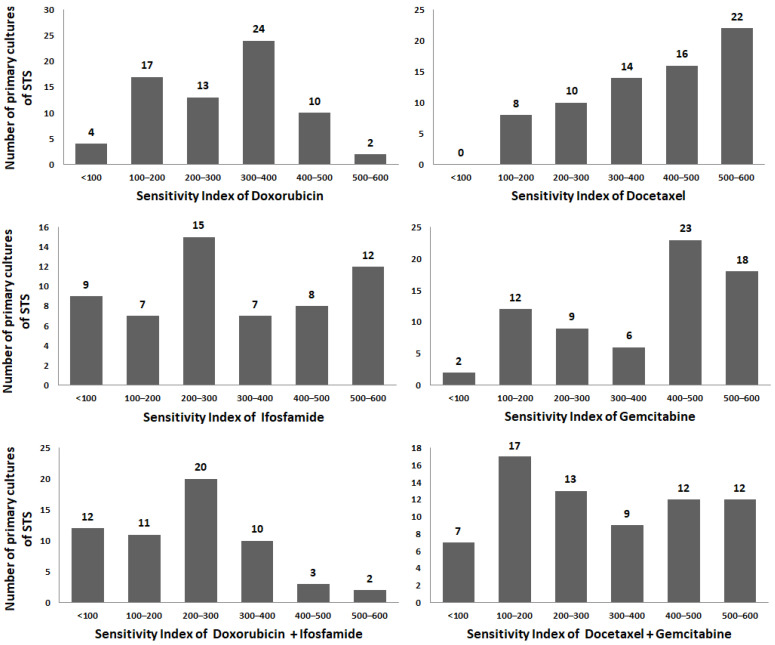
Frequency histograms of the sensitivity index for each single drug (Dox, Ifo, Doc, Gem) and two combinations (Dox + Ifo, Doc + Gem).

**Figure 3 ijms-23-03183-f003:**
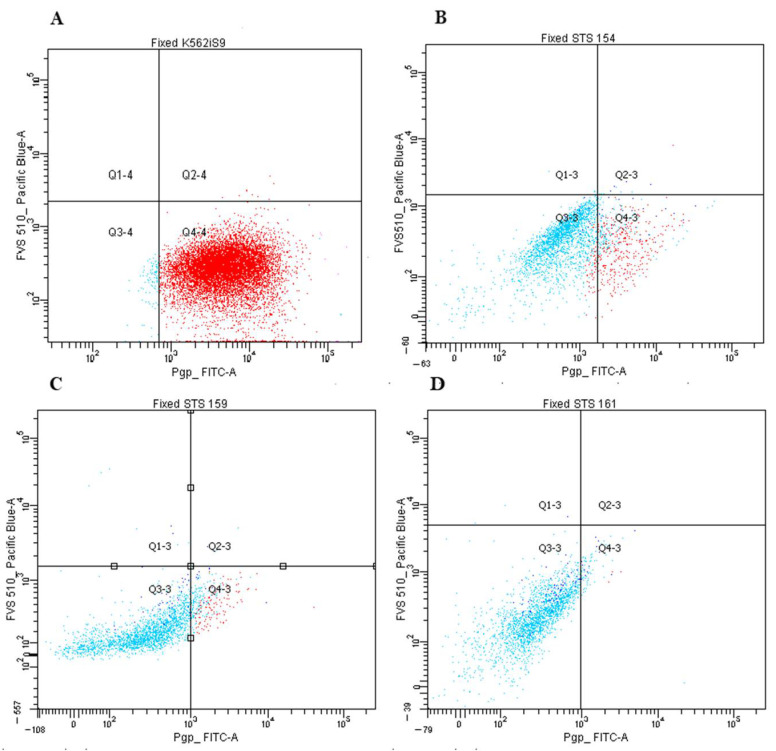
P-gp expression analysis by cytofluorometry. (**A**) K562/i-S9 cells were stained with FITC-labeled antibodies against Pgp (Pgp expressed by 98.2% cells); K562/i-S9 cell subline were obtained from chronic myeloid leukemia cell line K562 by transduction of *MDR1* (*ABCB1*). K562/i-S9 cells were used as positive control for the determination of Pgp staining. (**B**) Expression of Pgp in 33.8% of cells in STS 154 (synovial sarcoma). (**C**) Expression of Pgp in 8% of cells in STS 159 (leiomyosarcoma). (**D**) Expression of Pgp in 0.7% of cells in STS 161 (synovial sarcoma).

**Figure 4 ijms-23-03183-f004:**
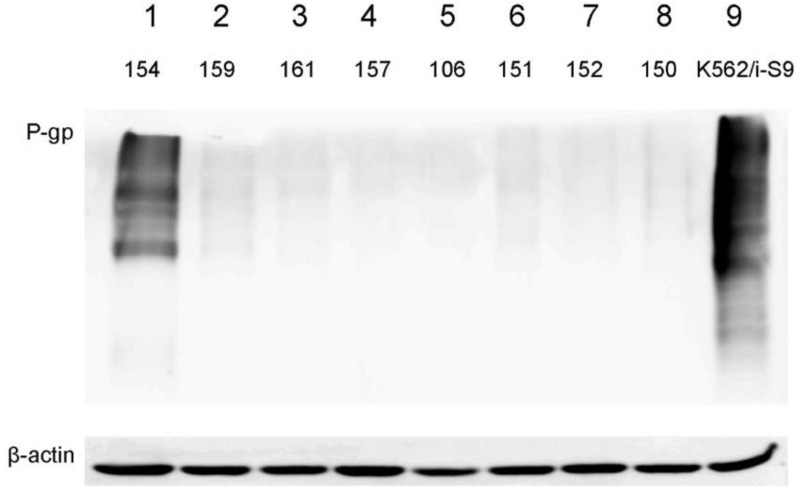
Validation of Pgp protein expression in STS samples by Western blotting. Lines 1–8: analyzed STS samples. Line 9: positive control of anti-Pgp antibodies activity (K562/i-S9 cells).

**Figure 5 ijms-23-03183-f005:**
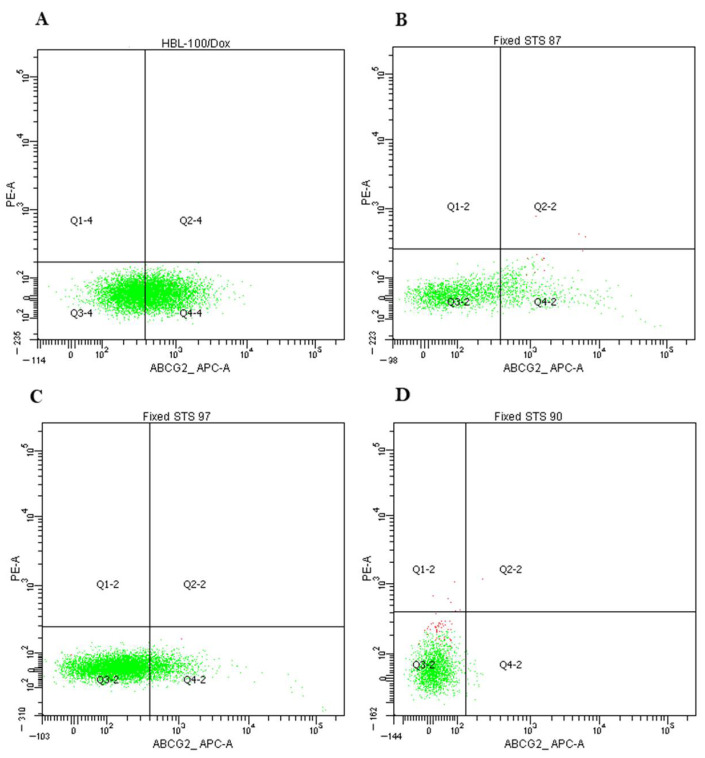
ABCG2 expression analysis by cytofluorometry. (**A**) HBL-100/Dox cells were stained with APC-labelled anti-ABCG2 antibodies (ABCG2 expressed by 50% cells). Dox-resistant cell subline HBL-100/Dox with overexpressed ABCG2 (BCRP) was obtained from the HBL-100 cells after the selection with Dox. HBL-100/Dox cells were used as positive control for the determination of ABCG2 staining. (**B**) Expression of ABCG2 in 17.1% of cells in STS 87 (liposarcoma). (**C**) Expression of ABCG2 in 8.6% of cells in STS 97 (undifferentiated pleomorphic sarcoma). (**D**) Expression of ABCG2 in 0.4% of cells in STS 90 (synovial sarcoma).

**Table 1 ijms-23-03183-t001:** Sensitivity index of STS primary cultures to studied drugs and its comparison to each other (Mann–Whitney test).

**Drugs**	**Dox**	**Ifo**	**Dox + Ifo**	**Doc**	**Gem**	**Doc + Gem**
	Sensitivity index (SI), mean ± SD					
	293 ± 121	315 ± 172	230 ± 134	402 ± 141	379 ± 162	304 ± 171
	** *p-* ** **value means**					
**Ifo**	**0.51**	**-**	**-**	**-**	-	-
**Dox + Ifo**	**0.0038**	**0.0065**	-	-	-	-
**Doc**	**<0.0001**	**0.0036**	**<0.0001**	-	-	-
**Gem**	**0.0006**	**0.04**	**<0.0001**	0.45	-	-
**Doc + Gem**	0.93	0.68	**0.03**	**0.0005**	**0.01**	-

**Table 2 ijms-23-03183-t002:** Association of the SI to drugs and clinical characteristics of patient and parameters of STS tumors (Mann–Whitney test), *р*-value.

Characteristics	n, *%*	Dox	Ifo	Dox + Ifo	Doc	Gem	Doс + Gem
SI, mean ± SD	70	293 ± 120	328 ± 167	238 ± 130	401 ± 145	374 ± 164	297 ± 172
Age							
<40 >40	23 (33%) 46 (67%)	**0.01 *** (higher resistance in >40 group)	0.41	0.10	0.08	0.08	0.17
Gender							
Female Male	33 (47%) 37 (53%)	0.49	0.54	0.82	0.70	0.89	0.16
Tumor size							
T1–T2 Т3–Т4	38 (54%) 32 (46%)	0.89	0.30	0.46	0.78	0.39	0.62
Histology types							
Undifferentiated pleomorphic sarcoma Liposarcoma Synovial sarcoma Others	24 (34%) 16 (23%) 11 (16%) 19 (27%)	**0.04 *** (Undifferentiated pleomorphic sarcoma more resistance)	0.74	0.62	0.12	0.34	0.13
Tumor grade							
G1–G2 G3	10 (15%) 56 (85%)	0.19	0.10	0.08	0.27	0.26	0.15
Newly diagnosed (without NeoCT) vs. Recurrent	22 (39%) 35 (61%)	0.11	0.59	0.18	0.63	0.92	0.86
Treatment characteristics							
Chemotherapy (all patient)							
No Yes	34 (49%) 36 (51%)	0.23	0.21	0.72	0.89	0.35	0.64
Newly diagnosed vs. Newly diagnosed with NeoCT	22 (59%) 15 (41%)	0.32	0.92	0.98	0.21	0.16	0.06
Tumor regression grade							
0–1 2–3	17 (73%) 6 (27%)	0.11	0.31	0.49	**0.0014 **** (higher resistance in 2–3 grade group)	0.25	0.06
Tumor regression grade after NeoCT							
0–1 2–3	10 (71%) 4 (29%)	0.053	0.29	0.55	**0.008 **** (higher resistance in 2–3 grade group)	0.053	**0.03 *** (higher resistance in 2–3 grade group)

NeoCT—neoadjuvant chemotherapy; * *p*-value < 0.05; ** *p*-value < 0.01.

**Table 3 ijms-23-03183-t003:** Correlation of expression levels of MDR-associated genes in STS samples (Spearman correlation): correlation coefficient (r)—gray background; *p*-value—white background.

	*YB-1*	*ABCB1*	*ABCC1*	*ABCG2*	*MVP*
*YB-1*	-	0.13	0.08	0.19	**0.36**
*ABCB1*	0.30	-	**0.67**	**0.88**	**−0.27**
*ABCC1*	0.55	**<0.0001**	-	**0.68**	0.07
*ABCG2*	0.10	**<0.0001**	**<0.0001**	-	−0.17
*MVP*	**0.001**	**0.02**	0.60	0.14	-

**Table 4 ijms-23-03183-t004:** Association of expression of MDR-associated genes and clinical characteristics of patient and parameters of STS tumors (Mann–Whitney test), *p*-value.

Characteristics	n, %	mRNA *ABCB1*	mRNA *ABCC1*	mRNA *ABCG2*	mRNA *YB-1*	mRNA *MVP*
Age						
<40 >40	23 (33%) 46 (67%)	0.06	0.22	**0.007 **** (high in >40)	0.48	0.54
Gender						
Female Male	33 (47%) 37 (53%)	0.10	0.16	0.34	0.20	0.52
Tumor size						
T1–T2 Т3–Т4	38 (54%) 32 (46%)	0.71	0.80	0.94	0.13	0.19
Histology types						
Undifferentiated pleomorphic sarcoma Liposarcoma Synovial sarcoma Others	24 (34%) 16 (23%) 11 (16%) 19 (27%)	0.07	0.12	0.23	**0.03 *** (high in Undifferentiated pleomorphic sarcoma)	0.39
Tumor grade						
G1–G2 G3	10 (16%) 56 (84%)	0.26	0.58	0.32	0.53	0.73
Newly diagnosed (without NeoCT) vs. Recurrent	22 (40%) 33 (60%)	0.47	0.74	0.21	0.94	**0.04 *** (low in recurrent)
Treatment characteristics						
Chemotherapy (all patient)						
No Yes	34 (49%) 36 (51%)	0.17	0.52	0.32	0.19	0.1
Newly diagnosed vs. Newly diagnosed with NeoCT	22 (59%) 15 (41%)	0.21	0.19	0.52	0.1	0.17
Tumor regression grade						
0–1 2–3	17 (73%) 6 (27%)	0.60	0.32	0.80	0.91	0.07
Tumor regression grade after NeoCT						
0–1 2–3	10 (71%) 4 (29%)	0.94	1	0.9	0.94	0.054

NeoCT—neoadjuvant chemotherapy; * *p*-value < 0.05; ** *p*-value < 0.01.

**Table 5 ijms-23-03183-t005:** Correlation of MDR-associated gene expression and SI of primary STS cell cultures in undifferentiated pleomorphic and synovial sarcomas (Spearman correlation).

Gene	n	Dox	Ifo	Dox + Ifo	Doc	Gem	Doс + Gem
	Undifferentiated pleomorphic sarcoma						
*ABCB1* (all patients)	24	ns	ns	ns	r = −0.51 ***p* = 0.009**	ns	r = −0.55 ***p* = 0.005**
*ABCB1* (recurrent patients)	6	ns	r = −0.90 ***p* = 0.03**	ns	r = −0.94 ***p* = 0.005**	ns	r = −0.77 *p* = 0.07
*ABCG2* (all patients)	23	ns	ns	ns	r = −0.48 ***p* = 0.02**	ns	r = −0.55 ***p* = 0.006**
*ABCG2* (recurrent patients)	5	r = −0.90 ***p* = 0.03**	ns	ns	r = −0.80 *p* = 0.10	ns	r = −0.90 ***p* = 0.03**
*MVP* (all patients)	22	ns	ns	ns	r = 0.49 ***p* = 0.02**	ns	r = 0.49 ***p* = 0.02**
*MVP* (patients with NeoCT)	8	ns	ns	ns	r = 0.69 *p* = 0.054	ns	r = 0.76 ***p* = 0.03**
	Synovial sarcoma						
*ABCC1* (all patients)	6	ns	r = 0.90 ***p* = 0.04**	r = 1.0 ***p* < 0.0001**	ns	ns	ns
*ABCG2* (all patients)	11	ns	ns	ns	r = 0.60 ***p* = 0.04**	ns	ns

NeoCT—neoadjuvant chemotherapy; ns—no significant.

**Table 6 ijms-23-03183-t006:** Association of Pgp and ABCG2 protein expression in STS primary cultures and clinical characteristics of patient and parameters of STS tumors (Mann–Whitney test).

Characteristics	n = 44, (%)	Pgp-Positive (n, %)	*p*-Value	n = 36, (%)	ABCG2-Positive (n, %)	*p*-Value
Gender						
Female Male	22 (50%) 22 (50%)	3 (14%) 2 (9%)	0.67	13 (36%) 23 (64%)	2 (15%) 13 (57%)	0.07
Age						
>40 <40	31 (70%) 13 (30%)	3 (10%) 2 (15%)	0.41	11 (36%) 25 (64%)	4 (36%) 11 (44%)	0.56
Tumor size						
Т1–Т2 Т3–Т4	28 (64%) 16 (36%)	3 (11%) 2 (13%)	0.83	20 (56%) 16 (44%)	9 (45%) 6 (38%)	0.43
Tumor grade						
G1–G2 G3	5 (13%) 34 (87%)	0 (0%) 5 (15%)	0.48	2 (6%) 34 (94%)	1 (50%) 13 (41%)	1.00
Histology types						
Undifferentiated pleomorphic sarcoma	16 (36%)	0 (0%)	**0.04** (higher in synovial sarcoma)	13 (36%)	5 (38%)	0.30
Liposarcoma	9 (20%)	1 (11%)		11 (31%)	7 (64%)	
Synovial sarcoma	6 (14%)	2 (33%)		8 (22%)	2 (25%)	
Others	13 (30%)	2 (15%)		4 (11%)	1 (25%)	
Treatment characteristics						
Chemotherapy						
No Yes	28 (68%) 14 (32%)	3 (14%) 2 (20%)	0.84	23 (64%) 12 (36%)	5 (22%) 9 (75%)	0.74
Tumor regression grade						
0–1 2–3	8 (89%) 1 (11%)	1 (13%) 0 (0%)	-	7 (47%) 8 (53%)	3 (43%) 3 (38%)	0.48

**Table 7 ijms-23-03183-t007:** List of mutations in ABC-transporters family.

Patient ID	Gender	Age	Histology Types	Gene	Reference Allele (Position)	Sequenced Allele	Mutations
STS 8	Male	37	Synovial sarcoma	*ABCA13*	T (chr7:48271819)	A	p.M718K
					A (chr7:48644634)	T	p.L4987F
STS 22	Male	37	Synovial sarcoma	*ABCC6*	T (chr16:16188976)	G	Intron mutation
STS 24	Male	62	Undifferentiated pleomorphic sarcoma	*ABCC6*	T (chr16:16188949)	G	p.H554P
					T (chr16:16188976)	G	Intron mutation
STS 29	Female	70	Undifferentiated pleomorphic sarcoma	*ABCB11*	C (chr2:168944742)	T	p.E825K
				*ABCC1*	G (chr16:16090474)	A	p.G844S
				*ABCC11*	C (chr16:48176983)	T	p.G1160D
STS 80	Female	43	Synovial sarcoma	*ABCA3*	C (chr16:2323599)	T	Nonsense Mutation
STS 93	Female	32	Undifferentiated pleomorphic sarcoma	*ABCG2*	C (chr4:88113437)	T	p.G354R
STS 104	Male	66	Undifferentiated pleomorphic sarcoma	*ABCA1*	C (chr9:104831048)	A	p.W590L
				*ABCA3*	G (chr16:2288133)	A	p.T908I
STS 106	Female	60	Undifferentiated pleomorphic sarcoma	*ABCC11*	T (chr16:48187226)	C	p.M970V
STS 119	Male	67	Undifferentiated pleomorphic sarcoma	*ABCC1*	C (chr16:16131855)	T	p.R1296W
				*ABCB1*	G (chr7:87536512)	T	p.N809K

**Table 8 ijms-23-03183-t008:** Drugs tested and their 100% TDC as used in the ex vivo tests.

Drug/Combination	100% TDC (mg/mL)
Doxorubicin	3.0
Ifosfamide (4-hydroxy-ifosfamide)	3.0
Doxorubicin + ifosfamide	3.0 + 3.0
Docetaxel	11.3
Gemcitabine	25.0
Docetaxel + gemcitabine	11.3 + 25.0

## Data Availability

The data presented in this study are available on request from the corresponding author. The data are not publicly available due to privacy restrictions.

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
