# Peer review of "Analysis of Multiple Drug Resistance Mechanism in Different Types of Soft Tissue Sarcomas: Assessment of the Expression of ABC-Transporters, MVP, YB-1, and Analysis of Their Correlation with Chemosensitivity of Cancer Cells"

_ijms, 2022, doi:10.3390/ijms23063183_

Round 1
Reviewer 1 Report
The authors Moiseeva end colleagues investigate the role of multidrug resistance (MDR) mechanism in different STS histotypes. In particular they assessed the expression level of a MDR associated genes panel in a STS case series of 70 patients. Moreover they established the matched patient-derived primary cultures which were characterized in terms of pharmacological profiling. Gene expression profiling on tumor lesion and pharmacological profile obtained from primary cultures were correlated in order to obtain a chemosensitivity index. The results provide evidences of both ABCB1 and ABCG2 as negative prognostic biomarkers and MVP as positive prognostic biomarker for chemosensitivity to taxane-based regimen in UPS. Moreover Pgp was found expressed in 5 out 44 STS samples especially in synovial sarcoma. In conclusion the authors reported that the study highlight the great heterogeneity of STS in chemosensitivity and that the ABC-transporters are not strictly involved in MDR mechanism onset.
The paper is interesting and has a good relevant to the field.
The manuscript would benefit from the following:
- The authors should report each histotype assessed. Moreover liposarcoma should be divided in WDLPS/ALT, DDLPS, MLPS and PLS.
- Tumor location should be specified.
- The authors should report which type of chemotherapy was administered in the neoadjuvant setting.
- The protocol number approved by Ethic Committee for that study should be included in the manuscript.
- Chemoresistance of UPS is well-known, in this regard other MDR associated genes including LAPTM4A and LAPTM4B have been already described and correlated to a lower chemosensitivty especially to anthracycline-base regimen. The role of these markers should be highlighted and the following works should be referenced: Primary Culture of Undifferentiated Pleomorphic Sarcoma: Molecular Characterization and Response to Anticancer Agents. Int J Mol Sci. 2017 Dec 8;18(12):2662. doi: 10.3390/ijms18122662. PMID: 29292724; PMCID: PMC5751264. Furthermore this recent work on the MDR in sarcoma should be included: Cellular plasticity and drug resistance in sarcoma. Life Sci. 2020 Dec 15;263:118589. doi: 10.1016/j.lfs.2020.118589. Epub 2020 Oct 15. PMID: 33069737.
- Limitations of the study should be reported.
Reviewer 2 Report
Please apply the following revisions:
1- Structure the Abstract (divide it in Sections: Background, Methods, Results, Conclusions)
2- Move this period:
‘’ We assessed the expression of ABC-transporters, MVP and YB-1 and analyzed their correlation with chemosensetivity of corresponding STS cells to the above mentioned chemotherapeutic agents. Drug resistance was estimated using in vitro chemosensetivity assay proposed by Kurbacher et al [18], which, nowadays, are widely used in many modifications [19, 20,21,22], based on different methods for estimation of cell viability [23, 24, 25].’’
From the Introduction Section to the Materials and Methods Section of the manuscript.
3- Add a Conclusions Section
Round 2
Reviewer 1 Report
The work has been improved. The authors have addressed some important issues raised. the paper could be considered for publication.
Author Response
Dear Peer Reviewer,
We are grateful very much for careful reading of our manuscript and valuable pieces of advance.
Reviewer 2 Report
Please apply the following revisions:
1- Move this period: ‘’ We assessed the expression of ABC-transporters, MVP and YB-1 and analyzed their correlation with chemosensetivity of corresponding STS cells to the above mentioned chemotherapeutic agents. Drug resistance was estimated using in vitro chemosensetivity assay proposed by Kurbacher et al [18], which, nowadays, are widely used in many modifications [19, 20,21,22], based on different methods for estimation of cell viability [23, 24, 25].’’ From the Introduction Section to the Materials and Methods Section of the manuscript. You can add it to the beginning of the Materials and Methods Section. In fact, it is important that the Methods Section starts with a general description of the Methods of the study.
2- In the Abstract, explain what 'Blokhin NMRCO' stand for. What is it? please make this clear for the reader
3- In the Abstract, explain what 'Doc' and 'gem' stand for. What are they? please make this clear for the reader
Author Response
Dear Peer Reviewer,
We appreciate your pieces of advance very much and have made all the required changes:
- We added the subsection “Methodological approaches” to the section “Material and Methods” and have moved the pointed out fragment of the Introduction (‘’We assessed the expression of ABC-transporters, MVP and YB-1 and analyzed their correlation with chemosensetivity of corresponding STS cells to the above mentioned chemotherapeutic agents. Drug resistance was estimated using in vitro chemosensetivity assay proposed by Kurbacher et al [18], which, nowadays, are widely used in many modifications [19, 20,21,22], based on different methods for estimation of cell viability [23, 24, 25].) to this subsection.
- We have provided the official name of our Cancer Research Center in the Abstract.
- We have added abbreviations of the chemotherapeutic drugs in the Abstract.
Round 3
Reviewer 2 Report
1- In the Abstract, please delete the name of the center from this sentence: ''STS specimens were obtained from patients of Blokhin National Medical Research Center of Oncology'' and replace it with information about the number of patients included in the study and the years (2018-2020).
2- Please add also the inclusion criteria for the patients to be enrolled in the study and the type of study: is it a retrospective study? is it a prospective study? Add this info briefly in the Abstract and in a more detailed way in the Methods Section of the Manuscript.
3- Change the title to a more generic description of the study, like:
''Analysis of multiple drug resistance mechanism in different types of soft tissue sarcomas: assessment of the expression of ABC-transporters, MVP, YB-1 and analysis of their correlation with chemosensitivity of cancer cells''
this is only an example of a possible clearer title, you can modify it staying on the same concept of giving a clearer glance to the reader,
for example:
Analysis of multiple drug resistance mechanism in different types of soft tissue sarcomas: a prospective study
or
Analysis of multiple drug resistance mechanism in different types of soft tissue sarcomas: a retrospective study
if the study was retrospective
Author Response
Dear Peer Reviewer,
- We added information about the number of patients and the years (2018-2020) of study.
- We added information about inclusion criteria and type of study in the Methods Section of the Manuscript.
- We would like to follow your suggestion to change the title to a more generic description of the study:
«Analysis of multiple drug resistance mechanism in different types of soft tissue sarcomas: assessment of the expression of ABC-transporters, MVP, YB-1 and analysis of their correlation with chemosensitivity of cancer cells».

Round 4
Reviewer 2 Report
Thank you for your revisions